# One-Class SVM-guided Negative Sampling for Enhanced Contrastive Learning

Dhruv Jain*[1], Tsiry Mayet[1], Romain Hérault[2], and Romain Modzelewski[1, 3]

[1]Normandie Univ. INSA Rouen LITIS 76801 Saint Etienne du Rouvray France
[2]Université Caen Normandie, ENSICAEN, CNRS, Normandie Univ, GREYC UMR6072, F-14000 Caen, France
[3]Nuclear Medicine Department Henri Becquerel Cancer Center Rouen France

## Abstract

Recent studies on contrastive learning have emphasized carefully sampling and mixing negative samples. This study introduces a novel and improved approach for generating synthetic negatives. We propose a new method using One-Class Support Vector Machine (OCSVM) to guide in the selection process before mixing named as **Mixing OCSVM negatives (MiOC)**. Our results show that our approach creates more meaningful embeddings, which lead to better classification performance. We implement our method using publicly available datasets (Imagenet100, Cifar10, Cifar100, Cinic10, and STL10). We observed that MiOC exhibit favorable performance compared to state-of-the-art methods across these datasets. By presenting a novel approach, this study emphasizes the exploration of alternative mixing techniques that expand the sampling space beyond the conventional confines of hard negatives produced by the ranking of the dot product.
The code is available here.

## 1 Introduction

Empirical evidence has demonstrated that unsupervised contrastive learning is a highly effective technique for acquiring high-quality features, making optimal use of a vast unlabeled dataset. It has gained considerable popularity as a pre-training strategy for a range of tasks such as classification, segmentation, and generative modeling like in [1–3]. Recent studies indicate that contrastive learning yields better performance than supervised learning [4, 5]. The core concept of contrastive learning is to bring similar features closer together in the feature space while highlighting the differences between dissimilar features. In this context, an "anchor or query" image embedding is intended to share similarities with a "positive or key" image embedding, while it is designed to be distinct from the "negative" image embedding ensuring a clear separation.

*Corresponding Author: dhruv.jain@insa-rouen.fr

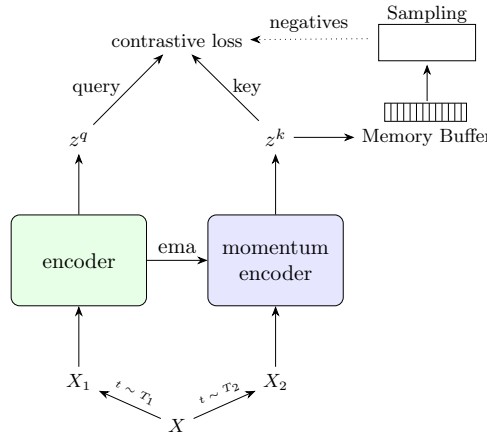

**Figure 1.** Visual illustration of the contrastive learning pipeline. In MoCov2 [5] the embeddings in the memory buffer/queue are used as negatives.

The selection process for positive and negative samples plays a crucial role in this domain, prompting continuous investigation into diverse methodologies. Momentum Contrast, or MoCo [6], is presented as a state-of-the-art baseline method in this paper utilizing two encoders: one for query and one for key. Instead of backpropagation, the key encoder's parameters are updated using a momentum-based method from the query encoder as shown in Figure 1. This causes the key encoder's parameters to change slowly, ensuring more consistent and stable representations of the negative samples. A dynamic dictionary of encoded data samples is constructed, functioning as a queue of negatives for contrastive learning. Typically, the positive pairs consist of different augmentations of the same image, whereas the negatives are sourced from distinct images. This paper investigates approaches to identify optimal negatives that could be interpolated and added to the existing queue to enhance the contrastive performance. There have been considerable efforts in identifying hard negative samples that are closely related to the query and hence harder to distinguish [7–9], however, there has been a lack of research dedicated to exploring different types of negative samples that are preferable for mixing. Focusing only on using hard negative samples for mixing can have a few issues:

Proceedings of the 6th Northern Lights Deep Learning Conference (NLDL), PMLR 265, 2025.

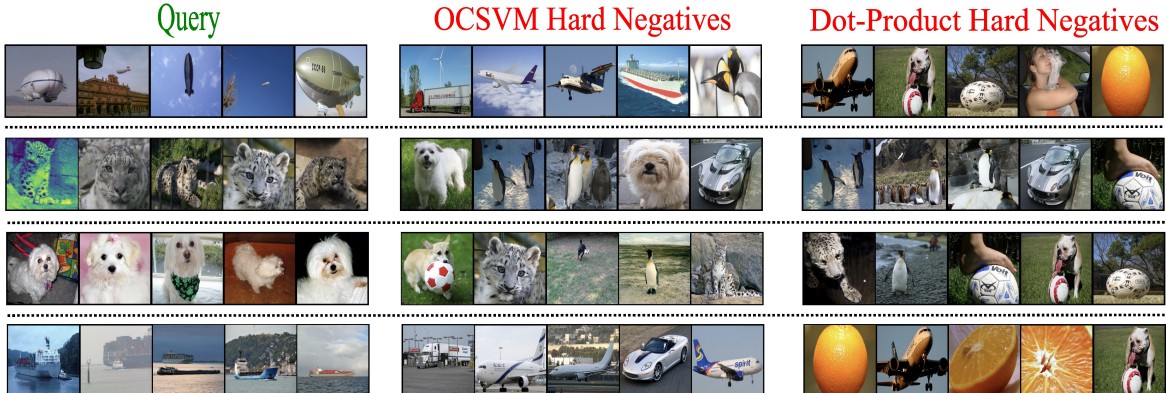

**Figure 2.** Samples of hard negatives by sorting dot-product (between query and negative sample) vs inliers identified by One-Class SVM (OCSVM). A set of 30 Query embeddings are selected for fitting the OCSVM and performing the dot-product on the Imagenet-10 dataset.

- Hard negatives might not encapsulate the broader patterns inherent in the data [10]. The synthetic negatives should possess the capacity to be non-redundant, in order to construct a more resilient representation.

- When engaging in the process of mixing, it is essential to construct harder negatives that are in proximity to the query [9]. Additionally, there should be a focus on accentuating the diversity among all negative samples to have a diverse and robust set of negatives.

Drawing inspiration from the aforementioned issues associated with negative mixing in contrastive learning, we present our approach :

- **Mixing OCSVM Negatives [MiOC]:** This method uses OCSVM to create new sets of synthetic negatives, assisting in the sampling of hard negatives. Figure 2 displays some examples of hard negatives found in the inlier region of the hypersphere produced by the OCSVM trained on 30 randomly chosen images of a certain class. It can be observed that the hard negatives given by OCSVM tend to be more similar to the query.

## 2   Related Works

### 2.1   Contrastive Learning

Contrastive Learning has emerged as one of the most effective strategies for self-supervised learning to acquire high-quality features before any downstream task. Here is a concise overview of the key improvements in contrastive learning. PIRL [11] was first introduced which was based on the notion that augmented images should have comparable features. Their findings demonstrated that their method could learn features from a discriminative task like solving a jigsaw. Another widely adopted approach SimCLR [12] generated positive samples by using two distinct encoders for different augmentations and creates negative samples from the remaining batch samples. This method required a large batch size to ensure a diverse set of negative samples for effective training. Contrastive Multiview Coding [13] was proposed that leverages the natural variations in data captured from different perspectives or modalities to learn more robust and generalizable representations. Momentum Contrast (MoCo) [6] was another approach that was proposed, which utilized a memory buffer as a queue to store negative samples and updated the weights of one of the encoders through momentum averaging, ensuring that the feature space does not exhibit significant disparities. Enhancement has been made to MoCo by several methods like MoCov2 [5], Relational Self Supervised Learning (ReSSL) [14] and Similarity Contrastive Estimation (SCE) [15]. A method described in [16] emphasized the importance of focusing on only the top 5% of the hardest negative samples to achieve optimal models. Additionally, the authors found that the most challenging 0.1% of negative samples are unnecessary and can hinder the training process in some cases, as they often consisted of pseudo-negative samples. There are some works like Student-t distribution with a neighbor consistency constraint(TNCC) and contrastive learning loss based on the Student-t distribution (CLT) [17] who introduced a novel loss that emphasizes prioritizing weak negatives over hard negatives. Alternative techniques have also been explored, such as [18, 19] which do not rely on negative samples.

### 2.2   Mixup

Several mixing approaches have enhanced the robustness of the learning process. MixCo [20] was based on the principle of understanding the rela-

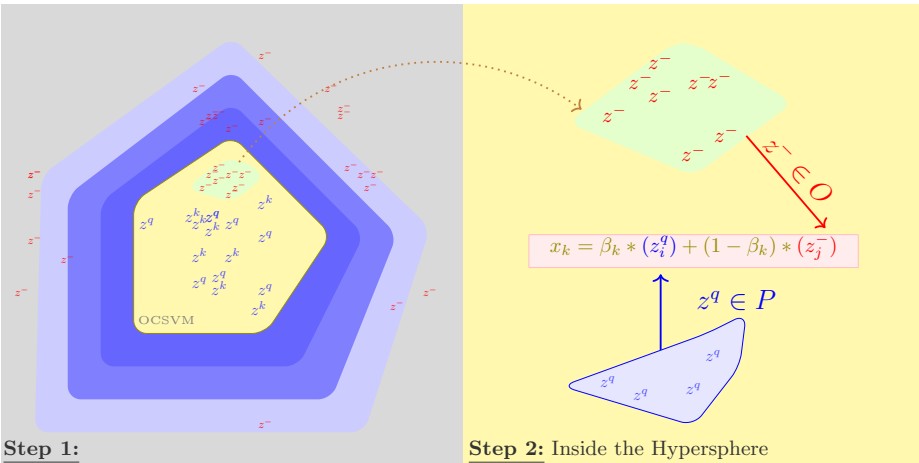

**Figure 3.** Illustration of our approach to create the synthetic set $S_o$: With every incoming batch, **Step 1**, OCSVM is trained on query $z^q$ and key $z^k$ belonging to a batch of embeddings to build the surrounding hypersphere. In **Step 2**, the inlier negative embeddings $z^-$ are randomly chosen and interpolated with a randomly chosen $z^q$. Here, P represents the set containing all $z^q$ in a batch, and O is the set of $z^-$ located within the OCSVM hypersphere (i.e., $z^q \in P$ and $z^- \in O$).$\beta$ is a hyperparameter which is randomly chosen between [0, 0.5]. *(Recommended to view in color)*

tive similarity between representations which indicated how much the mixed images retain the characteristics of the original samples. Another method, iMix [21], involved mixing images in a controlled manner, challenging the learning model to disentangle and identify the individual components of the mixed images. MoCHI [11] is another approach that generates two groups of synthetic negative samples. The first group is created by mixing hard negatives among themselves, while the second group is created by mixing hard negatives with the anchor. Another approach, named SynCo [22], introduced six strategies for generating diverse synthetic hard negatives in real-time.

## 3 Method

### 3.1 Principles of One Class SVM (OCSVM)

OCSVM can be considered as a method for class density estimation. These algorithms are widely employed in anomaly detection. OCSVM detects the smallest possible hyper-sphere that encompasses all the points belonging to a specific class [23, 24]. It can alternatively be viewed as a margin separator from the origin. The hypersphere is characterized by its center, c, and radius, r. The optimization problem can be expressed as follows:

$$\min_{r,c,\zeta} r^2 + \frac{1}{\nu n} \sum_{i=1}^{n} \zeta_i, \qquad (1)$$

subject to $\|\Phi(x_i) - c\|^2 \le r^2 + \zeta_i$ for all $i = 1, 2, \ldots, n,$

where $\Phi(.)$ is a non-linear transformation performed by the kernel function, $\nu$ is the tradeoff coefficient between the sphere volume and the outliers, and $\zeta_i$ are non-negative slack variables. After fitting the hypersphere to the data, any sample $s_i$ can be categorized into one of three groups: inner-sphere, outer-sphere, or boundary points. A functional form for the decision function, denoted as $f(s_i)$, is shown in Equation 2 to provide us with information about the orientation of $s_i$.

$$f(s_i) = \langle w, s_i \rangle - b - \rho, \qquad (2)$$

where $w$ is a normal vector to the hyperplane, $b$ is the bias term, and $\rho$ is the threshold. Here $f(s_i)$ can have one of the three ranges of values:

- $f(s_i) > 0$: $s_i$ is inside the decision boundary.
- $f(s_i) < 0$: $s_i$ is outside the decision boundary.
- $f(s_i) = 0$: $s_i$ is on the decision boundary.

The function $f(s_i)$ will be used to sample hard negatives, which are negative samples located near the query.

### 3.2 Our Proposition: MiOC

We propose to construct additional synthetic negatives (inspired by MoCHI [11]) by the linear interpolation of a randomly chosen query and a randomly chosen negative as shown in Equation 3.

$$\mathbf{x}_k = \frac{\tilde{\mathbf{x}}_k}{\|\tilde{\mathbf{x}}_k\|_2}, \text{ where } \tilde{\mathbf{x}}_k = \beta_k \mathbf{z}_i^q + (1 - \beta_k) \mathbf{z}_j^-, \quad (3)$$

Here, $\beta$ ranges from 0 to 0.5, interpolating a negative embedding $z_j^-$ with a query $z_i^q$. Two synthetic

**groups** of negatives $S_n$ and $S_o$ are created as shown in Figure 4. Each group consist of a number of synthetic negatives of type $\mathbf{x}_k$ from Equation 3. In the case of $S_n$, the negative is randomly chosen from the queue as described in Equation 4 and then interpolated with a randomly chosen query,

$$queue = \{s_i, \forall i \in [0...K]\}, \qquad (4)$$

where queue comprises of K samples in the negative memory buffer. $s_i$ is the $i^{th}$ negative sample. The second group of synthetic negatives, $S_o$, moves the **hard-negatives** closer to the query within the embedding space.

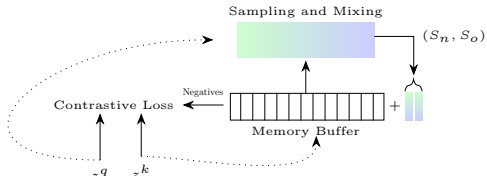

**Figure 4.** Illustration of the information flow in the sampling and mixing process for MiOC. The synthetic negatives are appended to the negative memory buffer and subsequently used for the contrastive loss.

OCSVM hypersphere. Subsequently, we search for the negative embeddings that fall within the bounds of the hypersphere to create the set $O$ by utilizing the decision function outlined in Equation 2. We use the InfoNCE loss as mentioned in MoCo [6] with our modification of the synthetic negatives as shown in Algorithm 1.

# 4 Experiments

We conducted reproducible experiments on Imagenet100, a subset of Imagenet1k [25]. Moreover, we conducted supplementary experiments to evaluate the overall performance of pre-training models under standard conditions, utilizing smaller datasets for linear evaluation.

## 4.1 Imagenet100

### 4.1.1 Experimental Setup

The training was conducted using a single Tesla A100. The images were resized to 224×224 and subjected to MoCov2 [5] augmentations. The pre-training and linear classification was done on the training set, while the results were reported on the validation set. The linear evaluation stage was conducted three times to show the standard deviation. We employed a MoCov2 [5] setup with a pre-training learning rate of 0.03 and a linear warm-up scheduler spanning ten epochs during which only $S_n$ negatives were generated. This allowed MiOC to include a certain number of samples within the hypersphere. Subsequently, a cosine scheduler was employed, and the pre-training process was conducted for 200 epochs using a ResNet50, which was trained from scratch. The embedding dimension and batch size were kept at 128. During the linear evaluation phase, we fixed the encoder, appended a linear layer on top, and conducted training for 60 epochs with a learning rate of 10 (as in [11]), employing a multistep scheduler with a factor of 0.1 at [30, 40, 50] epochs.

### 4.1.2 Result Analysis

The results for the linear evaluation on the Imagenet100 dataset are presented in Table 1. The

---

**Algorithm 1** Pseudocode for MiOC

**Require:**
   *img*: image from the loader,
   f_q and f_k: encoder networks for query and key,
   $C$: embedding dimension,
   *queue*: dictionary as a queue of K keys ($C \times K$),
   $t$: temperature,
   $O$: set of inlier ocsvm negatives,
   $f(s_i, hypersphere)$: returns orientation of $s_i$
   OCSVM: One-Class SVM
1: **for** each *image* in loader **do**
2:   $img_1 \leftarrow$ aug($img$) #augmented img
3:   $img_2 \leftarrow$ aug($img$) # another augmented image
4:   $z^q =$ f_q($img_1$) #queries: $N \times C$
5:   $z^k =$ f_k($img_2$) #keys: $N \times C$
     #Compute positive logits: N×1
6:   $l_{pos} \leftarrow$ bmm($z^q.view(N,1,C), z^k.view(N,C,1)$)
     #Obtaining the hypersphere parameters
7:   hypersphere = OCSVM(cat($z^q$, $z^k$))
     # Finding samples inside the hypersphere
8:   $O \leftarrow \{s_i, \forall i \in queue | f(s_i, hypersphere) > 0\}$
     #First set of synthetic negatives
9:   $S_n \leftarrow \{\tilde{x}_k = \beta_k z_i^q + (1 - \beta_k)z_j^- \mid z_j^- \in queue\}$
     #Second set of synthetic negatives
10:   $S_o \leftarrow \{\tilde{x}_k = \beta_k z_i^q + (1 - \beta_k)z_j^- \mid z_j^- \in O\}$
     #Concatenate the queue with the synthetic negatives
11:   $neg \leftarrow cat(queue, S_n, S_o)$
     #Compute negative logits: N×K
12:   $l_{neg} \leftarrow$ mm($z^q.view(N,C)$,
     $neg.view(C, K + len(S_n) + len(S_o))$)
     #Concatenate to calculate infonce loss
13:   $logits \leftarrow$ cat($[l_{pos}, l_{neg}], \dim = 1$)
14:   $labels \leftarrow \mathbf{0}_N$ #Initialize labels as zeros
     #Compute the loss
15:   $loss \leftarrow$ CrossEntropyLoss($logits/t, labels$)
16:   $loss$.backward() #backpropagate the loss
17: **end for**

**Notations:**
bmm: batch matrix multiplication;
mm: matrix multiplication;
cat: concatenation.

---

These hard-negatives are identified from the inlier negatives located inside the hypersphere that encompasses a batch of query $z^q$ and key $z^k$ embeddings as shown in Figure 3. We denote the set $O$ for these hard-negatives as in Equation 5.

$$O = \{s_i, \forall i \in [0...K] | f(s_i) > 0\}. \qquad (5)$$

To summarize a batch of (query ($z^q$) + key ($z^k$)) embeddings are used to train a high dimensional

Top1 % Accuracy and the k-NN scores have been compared for each model. k-Nearest Neighbour classifier predicts the data by considering the nearest neighbors based on features alone, without employing a linear layer. No training was necessary for this approach. We discovered that a value of 10 for "k" consistently performed the best across all models.

| Models | Imagenet100 | | | |
|---|---|---|---|---|
| | Acc Top1 % | k-NN | Effective Memory Buffer | Pretrain Time (Hrs) |
| CLT [17] | 68.17 | | | |
| TNCC [17] | 68.66 | | | |
| MoCo [6] | 73.4 | - | 16K | - |
| MoCo + iMix[21] | 74.2 | | | |
| CMC [13] | 75.7 | | | |
| CMC + iMix [21] | 75.9 | | | |
| SCE* [15] | 77.75 | 65.40 | | |
| MoCov2*[5] | $77.14_{\pm 0.24}$ | 65.10 | | 30 |
| | $77.32_{\pm 0.20}$ | 65.29 | | 31 |
| MoCov2 + MoCHI [1024, 512, 128]*[11] | $77.17_{\pm 0.06}$ | 63.73 | 17K | 40 |
| MoCov2 + SynCo*[22] | $77.15_{\pm 0.17}$ | 64.82 | | 27 |
| MoCov2 +MiOC[1024, 512]* | $\mathbf{78.07_{\pm 0.15}}$ | **65.89** | | 35 |

**Table 1.** Top1 % Accuracy on Imagenet100 for various models, with the effective memory buffer size (i.e., Queue-size + Synthetic Negatives). MiOC is represented with $[S_n, S_o]$ synthetic negatives respectively. * are implemented by us. Additional details about MoCHI [11] implementation and the hyperparameters can be found in the appendix.

MiOC demonstrated the best k-NN score and Top1 % Accuracy , while SCE [15] outperformed MoCoV2 [6] and MoCHI [11] in both metrics. SynCo [22] with a shorter pretraining time had lower k-NN and no significant improvement of Top1 % Accuracy than in MoCov2 [5]. We incorporated an expanded queue of 17K in MoCov2 [5]. Our findings demonstrated that the augmented queue length was not the primary factor contributing to the enhanced performance in MiOC. The computational load in MiOC mainly involved fitting the OCSVM and classifying points (inside or outside the hypersphere). This process is moderately resource-intensive when handling a small number of points, such as a batch of $(z^q + z^k)$. Despite the increased computation, it remains faster than MoCHI [11].

## 4.2 Linear Evaluation on Smaller Datasets

Assessing performance on smaller datasets provides insights into the model's capacity to generalize to new data. A strong performance on a small dataset implies that the model has acquired useful representations applicable across various tasks and datasets. We employed the pre-trained models trained on Imagenet100 for linear evaluation on four datasets- (Cifar10 [26], Cifar100 [26], STN10 [27], Cinic10 [28]). The initial three datasets are widely recognized as benchmark datasets, whereas Cinic10 [28] is a newly introduced dataset designed to serve as an intermediary between Cifar10 [26] and Imagenet [25]. The

| Models | Datasts(Top1 % Acc) | | | |
|---|---|---|---|---|
| | Cifar10 | Cifar100 | STL10 | Cinic10 |
| MoCov2 [5] | $80.24_{\pm 0.07}$ | $55.52_{\pm 0.18}$ | $73.58_{\pm 0.10}$ | $68.56_{\pm 0.05}$ |
| SCE [15] | $80.29_{\pm 0.05}$ | $55.50_{\pm 0.01}$ | $73.31_{\pm 0.01}$ | $68.59_{\pm 0.04}$ |
| MoCov2 +MoCHI[1024, 512, 128] [11] | $79.98_{\pm 0.03}$ | $54.79_{\pm 0.01}$ | $73.93_{\pm 0.03}$ | $69.12_{\pm 0.03}$ |
| MoCov2 +MiOC[1024, 512] | $\mathbf{81.01_{\pm 0.04}}$ | $\mathbf{56.27_{\pm 0.02}}$ | $\mathbf{74.36_{\pm 0.02}}$ | $\mathbf{69.40_{\pm 0.03}}$ |

**Table 2.** Comparison of linear evaluation performance on smaller datasets. Pretrained models from Imagenet100 (200 Epochs) were employed for the fine-tuning.

images were resized to 224×224 for Cifar10 [26], Cifar100 [26], Cinic10 [28], and 96×96 for STL10 [27]. We used a learning rate of 3 with a batch size of 128 and trained for 100 epochs with a multistep scheduler with a factor of 0.1 at [50, 70, 90] epochs. The output of the linear layer was adjusted according to the number of classes in each dataset. We can compare the results for linear evaluation on the smaller datasets in Table 2. MoCHI [11] outperformed MoCov2 [5] in both STL10 [27] and Cinic10 [28], whereas SCE [15] showed a slight improvement over MoCov2 [5] in Cifar10 [26], and Cinic10 [28], although the difference was not significant. MiOC demonstrated superior performance relative to all other models which clearly shows the benefit of our sampling and mixing strategy of negative embeddings. This insight sheds light on the importance of negative sample diversity and suggests that future research could explore more nuanced approaches to refine model performance further. Figure 5 exhibits the visualization of the ten

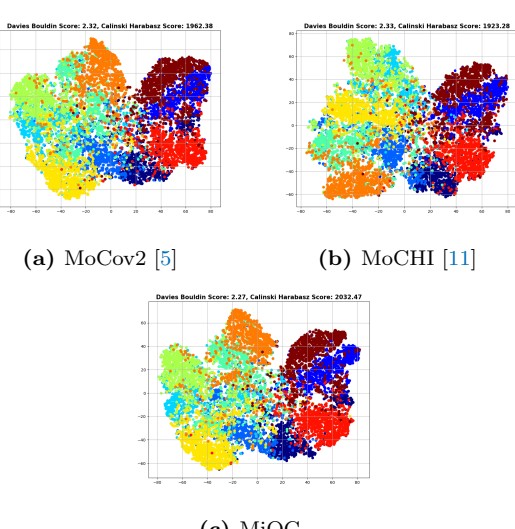

**(a)** MoCov2 [5]    **(b)** MoCHI [11]

**(c)** MiOC

**Figure 5.** Visualizing the linear evaluation by t-SNE and showcase ten classes of the Cifar10 [26] test set, revealing distinct clusters accompanied by **Davies Bouldin Score** (↓) and **Calinski Harabasz Score** (↑)

classes of the test set of Cifar10 [26] after performing linear evaluation on it, reduced to two dimensions using t-SNE. Additionally, it presents the Davies Bouldin Score and the Calinski Harabasz Score, both

metrics used to identify the optimal clustering for each model based on the features and labels. MiOC displays the lowest Davies Bouldin Score, and the highest Calinski Harabasz Score. Upon closer inspection, the t-SNE figure reveals that the distribution of the points in MiOC is better separated than in MoCov2 [6].

## 5    Future Work

In this paper, we introduced a technique for mixing negatives and proposed a novel approach to identifying hard negatives using One-Class SVM. Limited research has been conducted in this area, which opens up possibilities for exploring alternative sampling methods. The paper highlights the potential of OCSVM for a single application (image classification), although it may inspire other tasks. A potential area for future research could involve identifying and comparing the hard negatives selected by our method and ranking them based on the similarity of the dot product between the negatives and the query. Innovative ideas could be implemented on models like DINO [29], which does not utilize any negatives. Furthermore, it would be interesting to experiment with various anomaly detection methods to create synthetic negatives, such as those in [30] and [31], and compare their performance with MiOC.

## 6    Conclusion

In this article, we proposed a novel approach for mixing negatives that focuses on capturing the overall negative distribution rather than solely prioritizing hard negatives. Our method demonstrated a refined strategy for enhancing contrastive learning by integrating a broader spectrum of negative examples. Through testing on various datasets, our technique shows promise in outperforming some existing methods in multiple settings, highlighting the potential benefits of a negative sampling strategy. As the field progresses, we hope our work will contribute to the ongoing development of more sophisticated and effective learning algorithms.

## 7    Acknowledgement

This work was financially supported by the ANR Labcom LLisa ANR-20-LCV1-0009. We thank CRI-ANN, who provided us with the computation resources necessary for our experiments.

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

# A  Appendix

## A.1  Modified MoCHI

MoCHI [11] which is based on creating new sets of negative embeddings, i.e., $s_k$ and $s'_k$, by linear interpolation. (Equations using the same naming convention as in [11])

$$\mathbf{s}_k = \frac{\tilde{\mathbf{s}}_k}{\|\tilde{\mathbf{s}}_k\|_2}, \text{ where } \tilde{\mathbf{s}}_k = \alpha_k \mathbf{n}_i + (1 - \alpha_k)\,\mathbf{n}_j \quad (6)$$

Here, $\alpha$ represents a random variable ranging from 0 to 1. The variables $n_i$ and $n_j$ denote randomly selected hard negatives from the set N, which comprises hard negatives obtained by ranking the negative sample's dot product with a query. An additional set of more challenging negatives, denoted as $s'_k$, is generated using a similar method.

$$\mathbf{s}'_k = \frac{\tilde{\mathbf{s}}'_k}{\|\tilde{\mathbf{s}}'_k\|_2}, \text{ where } \tilde{\mathbf{s}}'_k = \beta_k \mathbf{q}_i + (1 - \beta_k)\,\mathbf{n}_j \quad (7)$$

Here, $\beta$ ranges from 0 to 0.5, interpolating the hard negative embedding $n_j$ with query $q_i$. The authors represent each model as $[N, s, s']$, where $N$ represents the number of hard negatives from which $s$ synthetic hard negatives and $s'$ synthetic harder negatives are derived. Although it was noticed that a higher value of N could lead to improved outcomes, the sorting of negatives was found to increase processing time. For each query they created the synthetic negatives which inturn increased the pretraining time and the effective queue-size. As the pretraining time increased to upto greater than 100 hours, we modified MoCHI [11] to perform interpolation on randomly chosen query and generate a total of $(s + s')$ synthetic negatives. Though this method is not comparable with the original MoCHI [11] method, it is closer to our method and shows the importance of sampling with OCSVM, hence we used it in our experiments.

## A.2  HyperParameter Selection for MiOC

We experimented with various settings for the different hyperparameter configurations for the Imagenet100 dataset. First, we conducted experiments with the OCSVM hyperparameters, including nu, gamma, and kernel, which significantly affect the hypersphere. We conducted the pre-training using a larger queue size of 65K to compare the pre-training time more efficiently. Table A.1 presents the Top 1% Accuracy associated with various selected hyperparameter combinations. We determined that the configuration with nu=0.1, gamma=0.1, and kernel=rbf yielded the best-performing optimized hypersphere. Interestingly, when employing identical values for nu and gamma, the linear kernel

exhibits slower performance than the RBF kernel with [nu=0.01, gamma=0.01]. Since we do not impose a maximum iteration constraint, in cases where the data lacks linear separability, the RBF kernel might demonstrate greater computational efficiency and converge more rapidly.

| Models | OCSVM Hyperparameters | | | Top1 % Acc | Pretrain Time (Hrs) |
|---|---|---|---|---|---|
| | nu | gamma | kernel | | |
| MoCov2 +MiOC[1024, 512] | 0.1 | 0.1 | rbf | **78.35** | 76 |
| | 0.01 | 0.1 | | 77.52 | 52 |
| | 0.1 | 0.01 | | 77.87 | 42 |
| | 0.01 | 0.01 | | 77.66 | 34 |
| | 0.01 | 0.01 | linear | 77.81 | 38 |

**Table A.1.** OCSVM hyperparameters, including nu, gamma, kernel, and their corresponding effects on Top1 % Accuracy.

We use the fastest ocsvm hyperparameters for all of the experiments, i.e., [nu=0.01, gamma=0.01, kernel=rbf]. Additionally, we carried out a study to explore the impact of various queue sizes during 100 and 200 pre-training epochs and present the linear evaluation results in Table A.2. For these experiments, we conducted all the pre-training anew while maintaining the Tmax of the cosine scheduler at the corresponding number of epochs. Using the 100-epoch pre-training model, MoCov2 [6] demonstrates reasonable performance and even surpasses MiOC with a 16K queue size. However, we believe that 100 epochs are insufficient to leverage the benefits of MiOC. However, at the 200-epoch mark, MiOC

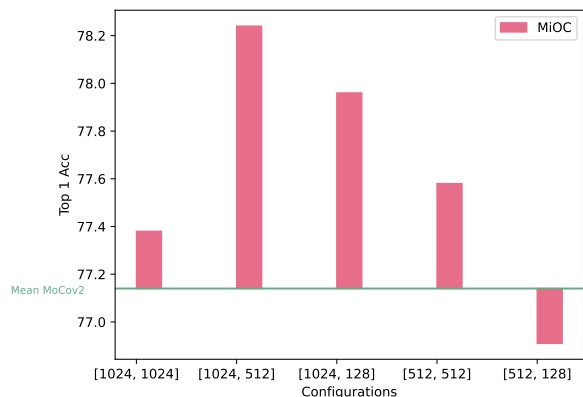

**Figure A.1.** Comparative Analysis of MiOC-$[S_n, S_o]$ hyperparameter optimization, showcasing Top1 % accuracy for each of the configuration pre-trained for 200 Epochs on Imagenet-100.

exhibits a clear advantage over other approaches. MiOC shows a slight improvement with a queue size of 65K, but adjusting the ocsvm's hyperparameters can lead to better results, as shown in the comparisons in Table A.1. We conducted a pretraining and linear evaluation for five distinct configurations for MiOC while maintaining the same hyperparameter settings as before.

The results for the optimal search of the best configuration are illustrated in Figure A.1. The green

| Models | Pretrained Epochs | Memory-Buffer Size | | | Pretrained Epochs | Memory-Buffer Size | | |
| --- | --- | --- | --- | --- | --- | --- | --- | --- |
| | | 4096 | 16384 | 65536 | | 4096 | 16384 | 65536 |
| MoCov2 [5] | 100 | 66.44 | **67.48** | 67.17 | 200 | 77.44 | 77.14 | 77.44 |
| MoCov2 +MoCHI[1024, 512, 128] [11] | | 65.97 | 64.98 | 66.91 | | 76.84 | 77.32 | 77.58 |
| MoCov2 +MiOC[1024, 512] | | **66.89** | 66.69 | **67.65** | | **77.81** | **78.07** | **77.66** |

**Table A.2.** This table displays the Top1 % accuracy achieved with different queue sizes (4096, 16384, 65536) across varying numbers of epochs (100 and 200). For both 100 and 200 epochs of model training, the same number of epochs was consistently used for the scheduler during pretraining.

| Models | Synthetic Negatives | | Acc Top1 % |
| --- | --- | --- | --- |
| | 1st Group | 2nd Group | |
| MiOC | $S_n$ | $S_n$ | 77.25 |
| | $M_n$ | $S_o$ | 77.23 |
| | $S_n$ | $S_o$ | 78.07 |

**Table A.3.** Summary of experimental results with different synthetic negative types and combinations.

reference line depicts MoCov2's [5] mean Top1 accuracy, highlighting the proposed model's performance improvement. We observed that [1024, 512] was the best configuration for MiOC. We tried more experiments with using different types of combinations of synthetic negatives as shown in Table A.3. Here $S_n$ and $S_o$ are the sets created as shown in Section 3.2. While we introduced a new set of synthetic negatives $M_n$ which uses *queue* as in Equation 4, though instead of mixing it with a randomly chosen query, we mix 2 negatives belonging to this set as in Equation 6. Here, we observe that the combination of $S_n$ and $S_o$ works the best and gives an advantage over MoCov2 [5].

## A.3 Linear Evaluation with Limited Data

We conducted further experiments wherein we restricted the number of samples per class to ranges between 10-1000 images for the Cifar10 [26] and Cinic10 [28] datasets. This approach is particularly useful in real-world situations where labeled data can be scarce or expensive to obtain. Linear evaluation with few images enables practitioners to use limited labeled data resources efficiently. Figure A.2 displays the results with limited training images. Our proposed techniques consistently demonstrate superior performance compared to other models. This experiment underscores MiOC's effectiveness for fine-tuning scenarios with limited data and showcases their adaptability. Notably, when the training set consists of only ten images per class, totaling 100 images, MoCHI's [11] performance is compromised, whereas MiOC consistently delivers comparatively stronger results. The performance of MiOC in such conditions presents promising opportunities for refining machine learning models for enhanced efficiency

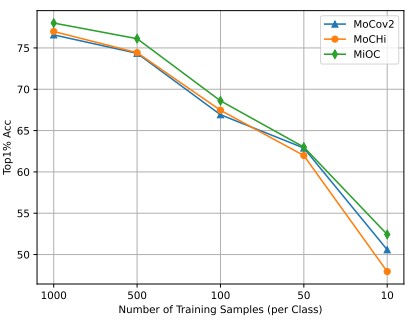

**(a)** Cifar10 [26]

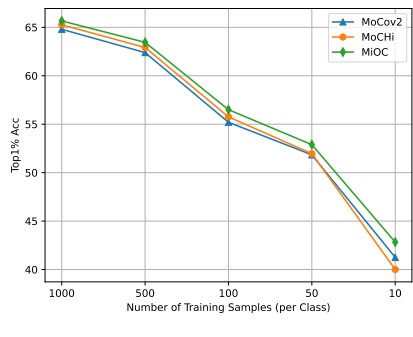

**(b)** Cinic10 [28]

**Figure A.2.** Comparison of Linear Evaluation under restricted training data. The X-axis depicts varying images per class utilized for training, while the Y-axis shows the Top-1% Accuracy.

in practical applications faced with data scarcity.

## A.4 Scalability for MiOC

We recognize the importance of the runtime for MiOC. We fit OCSVM to $z^k + z^q$ embeddings, which, with our batch size, amounts to $128 + 128 = 256$. Ideally, the algorithm can scale effectively up to a

| Batch Size | 128 | 256 | 512 | 1024 | 2048 | 4096 | 8K | 16K |
| --- | --- | --- | --- | --- | --- | --- | --- | --- |
| Time | 0.0017 | 0.0034 | 0.0109 | 0.0427 | 0.1691 | 0.6826 | 2.8508 | 16.7219 |

**Table A.4.** Comparison of batch size and the time required to fit OCSVM on a single batch

batch size of 4K. However, with larger batch sizes of over 8K, delays may become noticeable. We

show the OCSVM fitting for different batch sizes in Table A.4. Using a very large batch size (around 8K or 16K) can affect the step time. However, it is still feasible to use them for smaller/medium batches.

## A.5 MiOC Time Comparison

We compare the average batch execution time and batch size for the models listed in Table A.5, each evaluated over 100 batches. SynCO [22] demon-

| Models | Batch-Size | | |
|---|---|---|---|
| | 128 | 256 | 512 |
| SCE [15] | 0.34 | 0.38 | 0.83 |
| MoCov2 [5] | 0.34 | 0.42 | 0.46 |
| MoCHI [11] | 0.37 | 0.73 | 1.23 |
| SynCO [22] | 0.22 | 0.24 | 0.49 |
| MiOC | 0.41 | 0.61 | 0.81 |

**Table A.5.** Average time (in seconds) taken by 100 batches to run the specified model.

strates the fastest performance for smaller batch sizes (e.g., 128 and 256), but for larger batch sizes, its performance becomes comparable to MoCov2 [5]. MiOC, due to its additional OCSVM computation, requires slightly more time but scales efficiently with batch size. In contrast, MoCHI [11] shows a significant impact from increasing batch size.

