# OpenReview forum: "One-Class SVM-guided Negative Sampling for Enhanced Contrastive Learning"
_NLDL.org/2025/Conference — NLDL 2025 Oral_

### Official Review · Reviewer_a8Mf · 2024-09-26
**Review "One-Class SVM-guided Negative Sampling for Enhanced Contrastive Learning"**

**Confidence:** 3

**Summary:**

The paper "One-Class SVM-guided Negative Sampling for Enhanced Contrastive Learning" discusses a novel approach to generate negative samples for contrastive learning by utilizing the one-class SVM. Contrastive learning deals with the unsupervised learning of embeddings that can later be used for supervised down-stream tasks. In this context, for a given input $x$ (the anchor) a positive example $x^p$ and a set of negative examples $\{x^n_1,x^n_1\dots,\}$ are generated, and the learning task is to move the embeddings of the anchor and positive sample closer together, while increasing the embedding distance between the anchor and the negative samples. Empirically, it has been shown that a good set of negative samples is vital to contrastive learning (CL). To improve CL, the authors propose to use the one class SVM to detect if samples in the embedding space are close together (i.e. inside the SVM's circle) or far away (i.e. outside) and then use this information to construct new (difficult) negative examples. More formally, the authors generate two sets of negative samples: The first set is based on a random interpolation between the anchor and a continuously updated set of negative samples generated via the MiOC method. A second set of negative samples is generated by training a one class SVM on the current batch of anchor points and their negative samples. Points that are inside the SVMs decision boundary are deemed difficult (i.e. the SVM did not rate these points as outliers). These difficult samples are used to generate new negative examples by interpolation (this is similar to MiOC). The authors test their method on the ImageNet100K dataset and use the obtained embeddings for various down-stream tasks such as CIFAR10/100,Cinic10 and STL10. During experiments, the authors show that their OneClass SVM approach improves CL performance. Moreover, in a subsequent analysis they show that the negative samples generated by their approach have a higher diversity, which might explain the improved performance.

**Strengths:**

I think the overall idea of using the One-class SVM to generate better negative samples is smart and well-placed in the context of Contrastive Learning. The experiments clearly show the benefit of this method and, despite the comparably small datasets, I think this approach could merit further investigation by the community. Figure 3 helps to overall understanding and additional information in the appendix is also given to further explain some aspects of the method.

**Weaknesses:**

I mainly find two weaknesses with this paper, that, I believe, can be addressed during the rebuttal:

1) While I find the general method and intuition easy to follow, I had a difficult time understanding the exact method. First, some variables such as $s_k$ (eq. 3) are explained after they have been presented. Second, I am still a bit unclear on what data the SVM is exactly trained. I would appreciate a better explanation of the overall method / section 3.1. To be more concrete:
    a) On what data is the SVM trained?
    b) How is $\beta$ chosen exactly?
    c) What is the additional insight of using $k$-NN as well for the experiments?
    d) What is the exact connection of the data augmentation and the interpolation to generate the final embeddings? I understand that this is partly discussed in the MiOC paper, but I am not 100% sure how this is placed in this paper

2) The one-class SVM is known to be comparably slow, and you need to fit a new SVM for each data batch. While the authors mention the runtime during the experiments, clear numbers are missing. What is the impact in training the SVM? How much time does this cost per batch? How does it scale?

**Justification:**

I think the paper presents a novel method that might be of interest to other readers. I could not find obvious flaws in the method, experiments or presentation of the paper. While the paper itself (i.e. its presentation) can be improved, I don't think the paper's current stage warrants a rejection. Hence, I vote for acceptance.

---

> ### Author Rebuttal · Authors · 2024-10-24
>
> We would like to thank all reviewers for their thoughtful and constructive feedback on our paper. We appreciate the time and effort taken to read and evaluate our work, and we have carefully considered all suggestions to improve the quality of the manuscript.
> # Major Changes
>
> - Added Algorithm 1 for MiOC loss. (suggested by 5RzJ)
>
> - Added MoCo + iMix [21], SynCo [22] in Table 1. (suggested by 5RzJ)
>
> - Included standard deviation in Table 2. (suggested by 5RzJ)
>
> - Used sets as uppercase - $Q, S_o, S_n$, individual elements as lowercase: $x_k, s_i, z^q, z^-$ (suggested by 5y1S)
>
> - Changed $s_k \rightarrow x_k$ (suggested by 5y1S)
>
> - Added scalability analysis of MiOC in the appendix. (suggested by a8Mf)
>
> - Changed $O'\rightarrow O$
>
> # Detailed Response
>
> 1. We understand the need for a straightforward algorithm - “Algorithm 1" in the article which enhances readability and so is added in the article.
> We have simplified the variables.
>
> a. Overall Method (refer to Algorithm 1):
> - OCSVM is trained on a batch of query and key embeddings. (Step 7)
> - Then its decision function is used to determine if a negative sample is inside or outside the hypersphere. (Step 8)
> - Based on this the negatives are interpolated with the query. (Step 9/10)
> - Queue is appended with these synthetic negatives. (Step 11)
>
> b. Beta was chosen the same as in MoCHI[11].
>
> c. It can be important for feature representation to have good kNN performances for tasks like image retrieval as shown in DINO[29].
>
> d. There is no direct connection between data augmentation and interpolation. The interpolation adds extra negative samples to the queue without changing anything in key or query. While data augmentation is applied to query and key images as shown in Algorithm 1 (Step 2/3).
>
> 2. We recognize the importance of runtime analysis for MiOC.
>
>  $z^k + z^q$ embeddings with our batch size, amounts to $128 + 128 = 256$.
> Ideally, the algorithm can scale effectively up to a batch size of 4K.
> We show the OCSVM fitting time for different batch sizes:
>
> | **Batch-Size** |   128  |   256  |   512   |  1024  |  2048  |  4096  |   8K   |   16K   |
> |:--------------:|:------:|:------:|:-------:|:------:|:------:|:------:|:------:|:-------:|
> |  **Time(sec)** | 0.0017 | 0.0034 | 0.01098 | 0.0427 | 0.1691 | 0.6826 | 2.8508 | 16.7219 |
>
> Using very large batch sizes (around 8K or 16K) can affect the step time. However, it is still feasible for smaller/medium batch sizes.
> We have added this analysis to the updated article appendix.

---

### Official Review · Reviewer_5RzJ · 2024-10-05
**Interesting method, but a few concerns on originality and importance**

**Confidence:** 4

**Summary:**

The paper proposes to use one-class support-vector machine  (OCSVM) to guide the selection process for generating synthetic negatives for contrastive learning. The authors base their analysis on the idea of MoCHI
of interpolation between the randomly chosen query and a randomly chosen negative.

**Strengths:**

- Soundness/correctness: the model is sound and improves the performance upon the alternative strategy, MoCHI (although there are reservations as described in cons and the questions)
- Quality and clarity: the paper is well-written and clear (however, there are some concerns in questions below)

**Weaknesses:**

- Originality: in essence, the paper proposes a combination of two well-known approaches, MoCHI and OCSVM
- Importance: it seems to me that the proposed solution only marginally improves the performance, with the experimental results comparing the solution exclusively with MoCHI

Questions:
To expand upon the importance part, I suggest the following improvements:
- In this work, the authors focus their efforts on comparing the proposed method with MoCHI; it may be important to also present the comparison with other strategies for synthetic negatives generation, for example MixCo and iMix.
- It is also interesting that the results with MoCov2 + MoCHI do not improve upon MoCov2. Does it mean that the impact of MoCHI in this setting is actually negative? What would  be the reason for this, and could the same also be a problem for the proposed, closely linked, method?
- Should Davies-Bouldin score and Calinski-Harabasz scores be given with the confidence interval? The same comment applies to Table 2.
- The text around Eq 5-6 could be given as an algorithm to facilitate reading
- Figure 1’s samples of hard negatives do not (obviously) look convincingly better than the dot-product ones. I wonder if the authors could clarify
- (Minor question) Figure 2 comes before Figure 1, the authors need to fix it
- Given that the authors seem to be focus on OCSVM, I wonder if the alternative anomaly detection techniques could be used (e.g., Nizan & Tal, 2024; Guille-Escuret et al (2024)), and what would be their benefit?

Nizan & Tal, k-NNN: Nearest Neighbors of Neighbors for Anomaly Detection, WACV workshops, 2024
Guille-Escuret, C., Rodriguez, P., Vazquez, D., Mitliagkas, I., & Monteiro, J. (2024). CADet: Fully Self-Supervised Out-Of-Distribution Detection With Contrastive Learning. Advances in Neural Information Processing Systems, 36.

**Final Rebuttal Confidence:**

4

**Final Rebuttal Justification:**

I've checked the authors rebuttal, as well as discussion, and I think the authors addressed my comments and the comments from other reviewers. Therefore I suggest acceptance.

**Justification:**

In summary, the paper has a good idea, however the authors should clarify upon the analysis as presented above.

---

> ### Author Rebuttal · Authors · 2024-10-24
>
> We would like to thank all reviewers for their thoughtful and constructive feedback on our paper. We appreciate the time and effort taken to read and evaluate our work, and we have carefully considered all suggestions to improve the quality of the manuscript.
> # Major Changes
>
> - Added Algorithm 1 for MiOC loss. (suggested by 5RzJ)
>
> - Added MoCo + iMix [21], SynCo [22] in Table 1. (suggested by 5RzJ)
>
> - Included standard deviation in Table 2. (suggested by 5RzJ)
>
> - Used sets as uppercase - $Q, S_o, S_n$, individual elements as lowercase: $x_k, s_i, z^q, z^-$ (suggested by 5y1S)
>
> - Changed $s_k \rightarrow x_k$ (suggested by 5y1S)
>
> - Added scalability analysis of MiOC in the appendix. (suggested by a8Mf)
>
> - Changed $O'\rightarrow O$
>
> # Detailed Response
>
> - **Originality:** While OCSVM has been primarily used for anomaly detection, to the best of our knowledge, it has not been applied for sampling hard negatives or in the context of contrastive learning.
> This novel application constitutes a significant contribution to our work.
>
> - **Importance:**
> To reinforce the significance of our improved representations, we have provided comparisons of both top-1 accuracy and k-NN metrics. Additionally, our study includes comparisons between MoCov2 [5], SCE [15], and MiOC, offering a comprehensive evaluation of our method's performance.
>
> - We included MoCo + iMix [21], SynCo [22] in the ImageNet100 results in Table 1.
> MoCHI [11] and MiOC are the primary focus, as they involve comparing the synthetic negatives generated by the dot-product and OCSVM. iMix [22] and Mixco [20] are strategies that blend the query and key,  therefore we didn't compare them.
>
> - We reimplemented MoCHI [11] as the original code was not published. We found some limitations in their approach as detailed in Appendix A.1 and modified it accordingly.
> We note that other recent papers have also reported degraded performance of MoCHI [11] compared to MoCov2 [5]:
> 1. Giakoumoglou et Stathaki: SynCo Synthetic Hard Negatives in Contrastive Learning for Better
> Unsupervised Visual Representations (Oct 2024) Preprint~[22]
> 2. Ge et al.: Robust Contrastive Learning Using Negative Samples
> with Diminished Semantics (2021) Neurips
>
> The exact reason for the degraded performance is unclear, but it may be due to excessive reliance on hard negatives when generating synthetic negatives in contrast, MiOC takes into account all types of negatives.
> -  Davies Bouldin, Calinski-Harabasz scores are typically reported as single values without confidence intervals (e.g., in MixCo [20]). We have added confidence intervals for smaller datasets in Table 2 in the updated paper.
> - Thank you for the suggestion to add an algorithm. We have incorporated Algorithm 1 into our revised paper, as we believe it significantly enhances readability and overall comprehension.
> - This figure may be interpreted subjectively, as evidenced by contrasting opinions from different reviewers (5y1S found them convincing).
> However, we would like to highlight the first and last rows of the two types of hard negatives, which show more distinguishable images.
> - We have fixed it in the updated article.
> - Thank you again for bringing up this important point of exploring different anomaly detection methods. We have addressed it in the Future Work section of the revised paper. These approaches could potentially be compared with MiOC, and a new set of synthetic negatives could be explored.

---

### Official Review · Reviewer_pWvU · 2024-10-07
**Good read, sound approach, a few things could still be improved**

**Confidence:** 2

**Summary:**

The authors present MiOC, an approach for generating synthetic negatives to improve classification performance. They describe the method and evaluate MiOC using various data sets. The results show that it performs better than existing approaches.

**Strengths:**

The paper is very concise and gets to the point quickly. The language is very good (except for a few minor mistakes, see below). The approach appears to be sound, which is underlined by the evaluation results.

**Weaknesses:**

The paper does not have a dedicated section on related work. I suppose this is due to the fact that little research has been done in this area yet, but at the same time, this makes it hard to assess the paper's contribution. If there's anything the authors can do to improve on this, they should do it, as it would improve the paper's scientific quality! (e.g. not only compare quantitatively with existing approaches but also qualitatively or on a conceptual level).

Some remarks:

1 Introduction: The term "they" is used two times: "They utilise two encoders" and "They construct a dynamic dictionary". The authors should make it clear to what or whom 'they' refers in these cases.

1 Introduction: "[...] in the Figure 2" should be "[...] in Figure 2"

4 Experiments: The text switches back and forth between past and present tense (especially at the beginning). The authors should decide on a tense and then stick to it.

Appendix: I did not really understand how the appendix contributes to the paper. The only reference to it is in the caption of Table 1. Is the appendix really necessary? In particular, since MoCHI has already been described in a previous work [11]? The authors should make this clear and perhaps remove redundant information that has already been published.

**Final Rebuttal Confidence:**

2

**Final Rebuttal Justification:**

I already accepted the paper in my initial review, so my rating remains the same.

**Justification:**

Since I'm not an expert in AI theory, I mainly focused on the presentation. As far as I can tell, the approach is novel and contributes to the state of the art. Therefore, it is worth sharing. However, it could be that I did not understand every single detail. Also, I don't have a full overview of existing literature in this area. That's why I opted for "4 Accept" and not "5 Clear accept".

---

> ### Author Rebuttal · Authors · 2024-10-24
>
> We would like to thank all reviewers for their thoughtful and constructive feedback on our paper. We appreciate the time and effort taken to read and evaluate our work, and we have carefully considered all suggestions to improve the quality of the manuscript.
> # Major Changes
>
> - Added Algorithm 1 for MiOC loss. (suggested by 5RzJ)
>
> - Added MoCo + iMix [21], SynCo [22] in Table 1. (suggested by 5RzJ)
>
> - Included standard deviation in Table 2. (suggested by 5RzJ)
>
> - Used sets as uppercase - $Q, S_o, S_n$, individual elements as lowercase: $x_k, s_i, z^q, z^-$ (suggested by 5y1S)
>
> - Changed $s_k \rightarrow x_k$ (suggested by 5y1S)
>
> - Added scalability analysis of MiOC in the appendix. (suggested by a8Mf)
>
> - Changed $O'\rightarrow O$
>
> # Detailed Response
>
> - We thank you for this observation.
> We realize that Section 2 - Background may have been misinterpreted. It was intended to serve as the dedicated related work section.
> For clarity, we have renamed this section in the updated paper.
> We have corrected the remaining remarks in the updated paper.
>
> - Moreover, the appendix serves several purposes:
> 1. We reimplemented MoCHI [11] since the original code is not publicly available.
> To ensure reproducibility, we have included our implementation in the appendix.
> 2. It also includes hyperparameter search for MiOC, along with other experiments that may be of interest to researchers looking to understand its performance.

---

### Official Review · Reviewer_5y1S · 2024-10-10
**A simple but effective strategy for contrastive learning**

**Confidence:** 4

**Summary:**

The paper introduces a new method that mixes negatives selected with one-class SVM (OCSVM) for image-based contrastive learning. The method, called MiOC (Mixing OCSVM negatives), uses OCSVM to select inlier negative embeddings. These embeddings are combined with query embeddings, resulting in synthetic negatives. By using selected negatives, the method is able to generate better representations. For evaluation, authors have chosen image classification as the downstream task, using a larger dataset (ImageNet100) as well as smaller ones (CIFAR10, CIFAR100, STN10, CINIC10). In all cases, the proposed method has outperformed the baseline MoCov2 - even considering improved versions of it - and other SOTA methods.

**Strengths:**

The paper is clear, concise and well-illustrated. In particular, Fig. 1 properly shows a contrast of the proposed strategy and its counterpart dot-product hard negatives. The proposed strategy is simple but interesting. Results indicates that the strategy was effective for the problem.  Authors have indicated that the code will be available upon acceptation, which would be appreciated.

**Weaknesses:**

I only have two issues about the paper, most of my comments are minors. First, the k-NN strategy only appears in Sec. 4.1.2, but it is not clear the role of its analysis. In which way does it complement Acc Top1 %? What is supposed to be the conclusion of the acc top1 x k-NN comparison? Second, since the dot-product strategy is an important counterpart, a comparison of its results versus MiOC would improve the paper (or an explicit mention if it is already being done).

Other suggestions/questions:
[Eq 6] Is k_i a set? It is not clear, because it is the result of a set concatenation, but it is used in exp() as a unique embedding.
[Line 287] Include the references of the datasets.
[A.2] Which set was used for Hyper-parameter Selection?

Minors:
- All open quotes erroneously appear as close quotes.
- Use punctuation in equations and \nonindent after them if the text continues.
- Do not use 'x' for dimensions (as in 224x224). Use \times instead
- Figures 1 and 2 are in the wrong order in the paper. They should be indexed in the same order that they are referenced.
- [Line 114] "shortly" does not seem to fit in this sentence.
- [Line 132] Include the full form of CLT in the first occurrence.
- [Figure 3] Use math "belongs to" to indicate that z- and zq are in O' and P.
- [Lines 191-199] Use uppercase for sets and lowercase for elements to improve readability. Also, k is used as an index and as the size of set Q.
- [Line 257] between each model -> for each model
- [Line 200] comprises of all -> comprises all
- [Figure 5] visualising -> visualizing
- [Line 331] The paper highlights the potential of OCSVM for a single application (image classification), although it may inspire other tasks.
- Check references, since some of them do not have publication year.

**Justification:**

I mostly indicated minor suggestions, and the other issues do not significantly affect text quality. The strategy is simple and effective, and the problem is relevant and up-to-date.

---

> ### Author Rebuttal · Authors · 2024-10-24
>
> We would like to thank all reviewers for their thoughtful and constructive feedback on our paper. We appreciate the time and effort taken to read and evaluate our work, and we have carefully considered all suggestions to improve the quality of the manuscript.
> # Major Changes
>
> - Added Algorithm 1 for MiOC loss. (suggested by 5RzJ)
>
> - Added MoCo + iMix [21], SynCo [22] in Table 1. (suggested by 5RzJ)
>
> - Included standard deviation in Table 2. (suggested by 5RzJ)
>
> - Used sets as uppercase - $Q, S_o, S_n$, individual elements as lowercase: $x_k, s_i, z^q, z^-$ (suggested by 5y1S)
>
> - Changed $s_k \rightarrow x_k$ (suggested by 5y1S)
>
> - Added scalability analysis of MiOC in the appendix. (suggested by a8Mf)
>
> - Changed $O'\rightarrow O$
>
> # Detailed Response
> 1. We observed that the top1 \% accuracy from linear probing is sensitive to hyperparameters and presents a large variance in accuracy between runs when varying the hyperparameters similar to DINO[29].
> The k-NN approach offers several advantages: it does not require additional hyperparameter tuning or data augmentation, and it can be executed with only one pass over the downstream dataset, simplifying feature evaluation.
> 2. MoCHI[11] is a dot-product strategy in which we use hard negatives to create synthetic negatives.
> We have compared it with MiOC in Table 1 with Imagenet100 and Table 2 with smaller datasets.
> - Other Suggestions: Thank you for bringing this issue to our attention. $k_i$ was a typo. In the updated version, we have removed Equation 6, and instead, we have added an algorithm (Algorithm 1) of infoNCE for MiOC, which improves the article's readability. Line 287 - Dataset references have been added. In A.2, in the updated version, we have specified that hyper-parameter selection was conducted on ImageNet-100.
>
> - Minors: Thank you for bringing the minor errors in the manuscript to our attention. We have addressed and corrected them in the revised version.

---

### Meta-Review · Area_Chair_yuuB · 2024-11-02

**Recommendation:** Accept (Oral)
**Confidence:** 5

**Metareview:**

This paper presents MiOC, a novel method that leverages One-Class SVM (OCSVM) to improve contrastive learning through the generation of challenging synthetic negative samples. The approach is promising, showing competitive performance against other methods on several benchmark datasets. However, the paper would benefit from clearer articulation of its methodology and its placement within the existing literature. Additionally, more extensive comparisons to alternative sampling strategies and a deeper examination of the SVM’s computational costs would strengthen the study’s claims of efficiency and scalability.

While the approach shows potential, these improvements would enhance its accessibility and confirm the scalability of MiOC. Overall, the paper contributes a fresh perspective on negative sampling for contrastive learning, though more comprehensive validation is needed to maximise its impact.

**Suggested Changes To The Recommendation:**

1: I agree that the recommendation could be moved down

---

### Decision · Program_Chairs · 2024-11-06

**Decision:**

Accept (Oral)

**Comment:**

We recommend an oral and a poster presentation given the AC and reviewers recommendations.